# Can Infant Dyschezia Be a Suspect of Rectosigmoid Redundancy?

**DOI:** 10.3390/children9071097

**Published:** 2022-07-21

**Authors:** Carmine Noviello, Stefano Nobile, Mercedes Romano, Letizia Trotta, Alfonso Papparella

**Affiliations:** 1Pediatric Surgery Unit, Department of Woman, Child, General and Specialized Surgery, University of Campania “Luigi Vanvitelli”, 80138 Naples, Italy; mercedes.romano@libero.it (M.R.); letizia.trotta1992@gmail.com (L.T.); alfonso.papparella@unicampania.it (A.P.); 2Department of Woman, Child and Public Health, Fondazione Policlinico Universitario “A. Gemelli” IRCCS, 00168 Rome, Italy; stenob@iol.it

**Keywords:** infant dyschezia, redundancy, constipation

## Abstract

Infant dyschezia is a functional gastrointestinal disorder that occurs in children less than nine months of age. This disorder causes much anxiety among parents who consult different physicians when suspecting major intestinal problems. The aim of this study is to verify whether infant dyschezia involves an anatomic abnormality (redundancy) of the colon. In this retrospective study (48 months) we analyzed all the children younger than 9 months who came to our attention through the suspicion of gastrointestinal abnormality (Hirschsprung’s disease, anorectal malformations, colonic disorders or constipation). They all had a complete medical history, clinical examination and diagnostic tests, such as blood samples, suction rectal biopsy, a study of stool characteristics and, finally, a contrast enema. In cases with infant dyschezia, different colonic sizes and rectosigmoid length were measured, which created a ratio with the diameter of the second lumbar vertebra. These values were compared with those reported in the literature as normal for the age of one year. Of the 24 patients evaluated (mean age 4 months), 9 were excluded for different diagnoses (aganglionic megacolon, hypothyroidism, constipation). The comparison of the ratios obtained in the remaining 15 cases showed a significantly higher rectosigmoid length (redundancy) in children with dyschezia, 18.47 vs. 9.75 (*p* < 0.001). The rectosigmoid redundancy, a congenital anomaly already reported as a cause of refractory constipation, may be present in children with infant dyschezia.

## 1. Introduction

Infant dyschezia is a functional condition that occurs in a child under the age of nine months not suffering from constipation and who usually evacuates soft stools but sometimes has episodes with specific characteristics: straining and crying for a duration of at least 10 min before the successful or unsuccessful passage of soft stools (Rome criteria III and IV) [1,2]. According to the definition reported in the literature, it is a functional disorder that affects the gastrointestinal system, in fact, it is a combination of chronic and recurrent symptoms that does not depend on the anatomical, inflammatory or biohumoral anomalies of the bowel. Statistical data on the prevalence and natural history of infant dyschezia are not well reported in the literature [3,4,5]. There is no specific treatment for this disorder, but the most important thing is to reassure parents about the functional nature of the problem and the benign evolution over the months.

However, this discomfort can occur several times a day and, despite the reassurances of pediatricians, it creates a lot of anxiety in parents who consult different specialists and allow children to undergo continuous medical examinations, as well as diagnostic tests, which in some cases are also invasive tests. Furthermore, it must be considered that some intestinal pathologies, especially in the first months of life, may occur with the non-specific disorders of evacuation and crying.

The aim of this study is to verify whether an intestinal anatomical problem (redundancy of the sigmoid colon), already found in pediatric patients as a cause of severe constipation refractory to medical treatment and sometimes the cause of sigmoid colon volvulus, occurs in infants with dyschezia. An early and accurate diagnosis of the anatomical abnormality is important to suspect acute problems and to adequately treat constipation in order to prevent severe forms [6].

## 2. Materials and Methods

This retrospective study was conducted over a period of 4 years, between January 2018 and December 2021, and included all infants, aged less than 9 months, referred to the attention of our Pediatric Surgery Unit suspected surgical problems. The most important issues to suspect were Hirschsprung’s disease (HD), the anatomical malformations of the colon, minor recto-anal malformations and forms of constipation related to organ dysfunctions (thyroid, liver, kidneys).

Clinically, the suspicion of surgical pathologies was due to the need for rectal stimulation for evacuation, abdominal distension and/or other non-specific disorders during bowel movements. In the general classification of the patient, in each case careful anamnesis has been carried out with special attention to the gestational age, birth, the first passage of meconium, the daily diet (type of milk, weaning age, use of fecal softeners), the use of suppositories or anal stimulation. Then, the patient had a careful general clinical examination, with the particular evaluation of the abdomen and the anorectal region to exclude malformations involving this district and presence of anal fissures. Hegar probes were used to assess the presence of small anal malformations: the diagnosis of anal stenosis in case of the difficult introduction of the Hegar probe number 8. The infants were also subjected to general and specific hematological tests (exams were conducted to rule out hypothyroidism, liver and kidney disease).

The day after admission to the hospital, two blinded operators separately assessed the characteristics of the feces (hard, pasty, soft, mucousy or watery) of all infants for 3 consecutive bowel movements, and then the evaluation forms were compared in order to have an objective evaluation. Subsequently, a radiological examination was carried out to study the anatomy of the colon: a contrast enema (CE) was performed without bowel preparation before the examination in order to avoid emptying the filled intestinal tracts of feces. To rule out colonic aganglionosis, patients underwent rectal suction biopsies (RSB). For the purpose of the study, infant dyschezia was diagnosed in the cases of negative blood tests, negative RSB, normal physical examination and soft stools during bowel movements. In these cases, the CE images were studied, then the patients followed a pediatric follow-up until one year of age.

### 2.1. Contrast Enema Procedure

For the CE, we used as a contrast (Iopamide) diluted with sterile solution (ratio 1:4), which was perfused through a 6 Ch rectal probe fixed to the anus but without a balloon; the contrast was allowed to flow out of a collection bag positioned 50 cm above the level of the patient’s bed and the perfusion was stopped when the contrast reached the ascending colon.

### 2.2. Rectal Suction Biopsy Procedure

RSBs were performed on the patient in the supine position without sedation by a surgeon and a nurse. By using Noblett’s instrument, 3 biopsies were performed in three different sites at a distance of 3 cm from the dentate line (two laterally, one on the posterior margin). One of the biopsies was fixed in 4% formaldehyde, the others were immediately sent for the study of the acetylcholinesterase (AChE) activity and Calretinin staining. Ganglion cells were identified by hematoxylin and eosin staining. Biopsies were considered positive for HD in the case of absence of the ganglion cells and increased AChE activity or negativity for calretinin study. In non-diagnostic cases (inadequate sample or dubious results for calretinin and AChE activity), a surgical rectal biopsy was performed.

### 2.3. Study Design

To obtain better results, three blinded operators (two pediatric surgeons and a pediatrician) independently measured the size of the colon on the same radiographic image, evaluating the diameters of the rectum, sigmoid colon, descending colon, transverse colon and ascending colon, then measuring the length of the sigmoid colon (Figure 1). To standardize the results as much as possible and avoid weight and age-related variations, the ratios between the measured diameters of the various intestinal segments and the length of the sigmoid colon with the width of the body of the second lumbar vertebra (L2) were calculated. This same procedure has already been performed and reported by other authors as well as in our previous study of children with severe constipation older than 1 year [6]. The calculated ratios were compared with those reported as normal in one-year-old children in the literature [7]. The distribution of the data was verified by the Shapiro–Wilk test on median values with an interquartile range. In comparison, the study data used as a control was obtained with the Mann–Whitney U test. The level of the p-value indicating significance was set at 0.05. The Social Science Statistical Package (SPSS) v.22 (IBM, Armonk, NY, USA) was used to perform statistical analyzes.

## 3. Results

During this 48-month study period, 24 children were referred to our attention because of suspicion of surgical malformations: aganglionic megacolon, anorectal malformation, colonic malformation or the presence of non-functional constipation. The main reason for referring the patient was a non-specific problem with passing stool (thirteen children), while six cases had recurrent abdominal distension associated with crying and five reported bowel movements only with rectal stimulation. The mean age of the patients (who came to our attention) was 4 months (range 1 month–9 months).

### 3.1. Anamnestic Data

All the patients were the subjects of a careful anamnesis, which revealed that breastfeeding was present in six infants (25%), mixed feeding (breast and formula milk) in ten (42%), and the remaining eight (33%) were fed artificial milk, while fifteen had been weaned. The treatment for constipation was also studied: five children (21%) used fecal softeners, three suppositories, and six (25%) used rectal stimulation during crying and episodes of pain. The data concerning the birth did not show prematurity, while the passage of the first meconium occurred within 24 h in sixteen cases (67%), within 48 h in seven cases and only in one in 72 h.

### 3.2. Clinical Data

Clinical examination found one case (8-month-old infant) with anal fissures and a minimal distended abdomen in five cases. A Hegar rectal probe was easily used in all cases without finding anal stenoses, and only caused pain in the child with anal fissures. In all 24 cases, blood samples were collected, and the results revealed one case of hypothyroidism that was referred to the endocrinologist and excluded from the study.

### 3.3. Investigation Data

CE was performed in 23 cases (excluding the patient diagnosed with hypothyroidism) without bowel cleansing prior to examination. The diagnosis of HD was made in six cases (mean age 2 months) for RSB positive: the absence of ganglion cells in all cases, the increased AChE activity in five cases and negative calretinin in all cases. Surgical biopsies were not required in any cases. Five infants with HD used anal probe stimulation and three showed a difference in colonic caliber on CE. These six patients were excluded from the study and underwent surgical treatment (laparoscopic biopsies and laparo-assisted endorectal Pull-Through). Two other patients (age of the two cases 7 and 9 months) were excluded from the study after assessing stool characteristics (hard stool). Other characteristics of the stools were: pasty stools in three cases, soft stools in fourteen cases (two used intestinal softeners), watery stools in five cases (three used fecal softeners), and mucous stools in no cases.

### 3.4. Study Data

At the end of the examination, the diagnosis of infant dyschezia was confirmed in fifteen cases (nine males) who were recruited for the study. The mean age at presentation was 6 months (range: 2–9 months). In all patients, growth was regular for age, while the duration of symptoms was different in patients; on average, 5 months (range 1 to 7 months). The characteristics of the stools in these children showed fourteen soft stools and one pasty stool. Then their CE images were studied: first the diameters of the various portions of the colon and the length of the sigmoid colon were measured, then the ratios with the diameter of the second lumbar vertebra (L2) were calculated. The ratios calculated by the three blind operators were compared and an average was created to compare with the data reported in the literature. Comparison with the control ratios showed that the rectal diameter was similar in our patients and those reported by the other authors (1.46 vs. 1.27), as was the sigmoid diameter (1.20 vs. 1.05), the diameter of the descending colon (1.01 vs. 1.15), the diameter of the transverse colon (1.28 vs. 1.33) and the diameter of the ascending colon (1.24 vs. 1.33). The only value that appears significantly higher (*p*-value < 0.001) is the length of the rectosigmoid tract (18.37 vs. 9.81). (Table 1). Patients diagnosed with infant dyschezia had, in every case, a follow-up until one year of age: all showed the resolution of the problem and normal evacuation for age.

## 4. Discussion

Until a few years ago the criteria for diagnosing infant dyschezia were: “at least 10 min of effort and crying before the successful passage of soft stools in a child who usually evacuated normal and soft stools under the age of six months” [1,8]. However, over the last few years and with the analysis of statistical data on the prevalence and resolution of the disorder with the passing of the months, the authors noted that infant dyschezia was present even after 6 months of life and therefore the age of inclusion has recently been extended from 6 months to 9 months [2,9]. In addition to the age-related problem of the disorder, the concept that it is a functional disorder of an infant who is “otherwise healthy” is very important, according to the definition of the term. No treatment is necessary, but it is important to reassure the parents that this is a functional disorder due to impaired coordination between abdominal pressure and the relaxation of the anal sphincter and that it will improve over the months.

However, it should be considered that certain intestinal malformations in infants, even major pathologies, such as low anorectal malformations, Hirschsprung’s disease, colonic stenosis, hypothyroidism, renal and hepatic dysfunctions, may present, in the infant, non-specific alterations of evacuation, abdominal distention or crying. If we consider that infant dyschezia can occur several times a day, with the child crying and becoming red in the face, the disorder creates huge stress in the parents, leading them into a vicious circle: despite being reassured by the pediatrician about the lack of treatment and the positive evolution of the disorder, they tend to consult other physicians, who are not always able to make the diagnosis; therefore, in some cases, they recommend treatments to facilitate evacuations, like rectal stimulation, suppositories and/or fecal softeners [10].

These remedies will obviously not have significant outcomes on episodes of crying and pain, therefore parents will consult other doctors, who, evaluating a clinical state that is no more characteristic and believing they are underestimating a pathology that needs treatment, will send the children to specialists who will perform diagnostic tests, including invasive tests, such as contrast enemas and rectal suction biopsies. By evaluating our series of 24 cases with suspected surgical disease, we can see that 14 of them (58%) used treatments to help with bowel movements (five used fecal softeners, three suppositories and six rectal probe stimulation).

When these children who are treated for constipation are examined a pediatric surgeon, it becomes difficult to exclude malformations without conducting diagnostic tests. In our study, an anamnestic, clinical, hematological examination and a study of the characteristics of the stool were performed on 24 infants sent for a suspicion of intestinal organic disorders, which allowed the diagnosis of three cases of constipation, one with anal fissures and one related to thyroid dysfunction. Of the remaining 21 cases, the use of contrast enemas diagnosed three differences in colon caliber; moreover, rectal suction biopsies permitted the diagnosis of Hirschsprung’s disease in the aforementioned three patients, plus three others who appeared normal to the CE. When analyzing the cases of infant dyschezia, we noted that the only difference was the length of the symptoms: some were referred shortly after the onset of the disturbances, others after months. This is probably also linked to parental compliance. In our case series, we did not assess all infant dyschezia present during the study period, but those that were referred to our center for suspected intestinal malformation, leading to select a group of patients with special features; in fact, after our analysis, the 15 infants had a sure diagnosis, as we ruled out other pathologies.

The ratios calculated in our patients were compared to those reported in the literature as normal for the age of 1 year. In fact, these authors measured the same parts of the colon in patients who came to their observation because of intestinal intussusception [7], therefore without chronic problems of evacuation. The comparison between the various ratios shows a precise correspondence for all values except for the rectosigmoid length: in our 15 cases there seems to be a longer rectosigmoid length than in the group reported as control. This abnormality known as the rectosigmoid redundancy of the colon has already been reported by us in elderly patients with severe constipation.

For the purpose of this study, we measured the size of the colon on contrast enema images in 15 patients who received the final diagnosis of infant dyschezia. To make the result more homogeneous, so that it was not influenced by the age and weight of the patient, we measured the diameters of the rectum, sigmoid colon, descending colon, transverse colon and ascending colon, then the rectosigmoid length and created a relationship with the width of the second lumbar vertebra. The measurements were performed by three blind operators, but on the same image of the contrast enema, then an average was obtained between the three measurements. This method was also used in the control study [7] and in our previous article on redundancy in patients with constipation [6]. The aim is to obtain better results and to be able to compare them without fear of operator errors. When analyzing the measurements made, however, we saw that the differences between the various operators were all less than one centimeter, before creating the ratio with L2.

The true significance of the redundancy of the rectosigmoid tract of the colon is unclear. We know that the colon has the function of collecting feces and reabsorbing liquids, so in cases where this intestinal tract is longer, we can hypothesize that there is a greater reabsorption of liquids and a greater collection of feces, leading to hard and abundant stools.

We believe it is important to recognize the presence of a redundancy of the rectosigmoid colon, because this anomaly can rarely, and especially in some ethnic groups, predispose to acute episodes of volvulus [11,12,13,14,15,16]. Further studies are needed to understand the treatment of this abnormality. Having a longer part of the colon leads to constipation over the years, probably because the redundant pathway corresponds to a slow passage of stool [17,18]; in our opinion only adequate medical therapy, associated with a correct intake of fluids, can avoid severe forms of constipation in adults, which some authors believe need to be treated surgically [19] up to colectomy [20,21].

The presence of rectosigmoid redundancy does not change the benign nature of infant dyschezia, which, however, is due to a lack of coordination between the anal sphincter and the progression of the contents of the colon. In our opinion, newborns experiencing these symptoms, which tend to resolve after nine months, have to be monitored over time to check for the onset of constipation and adequate treatment must be given to avoid the issue reaching severe and medical refractory forms.

## 5. Conclusions

In conclusion, our study shows that rectosigmoid redundancy, a congenital condition that can, though rarely, predispose acute conditions (volvulus of the sigmoid colon), but which, over the years, frequently causes severe and refractory to medical treatment constipation) may show early signs, such as infant dyschezia, due to the difficulty that the redundant tract creates in the passage of gas and feces through the sigmoid colon. Therefore, we believe that children who present this disorder in the first few months, which tends to be resolved with growth, should be followed up even when the dyschezia is resolved, as they may later develop a severe form of constipation that needs to be treated quickly to avoid other colon deformities.

## Figures and Tables

**Figure 1 children-09-01097-f001:**
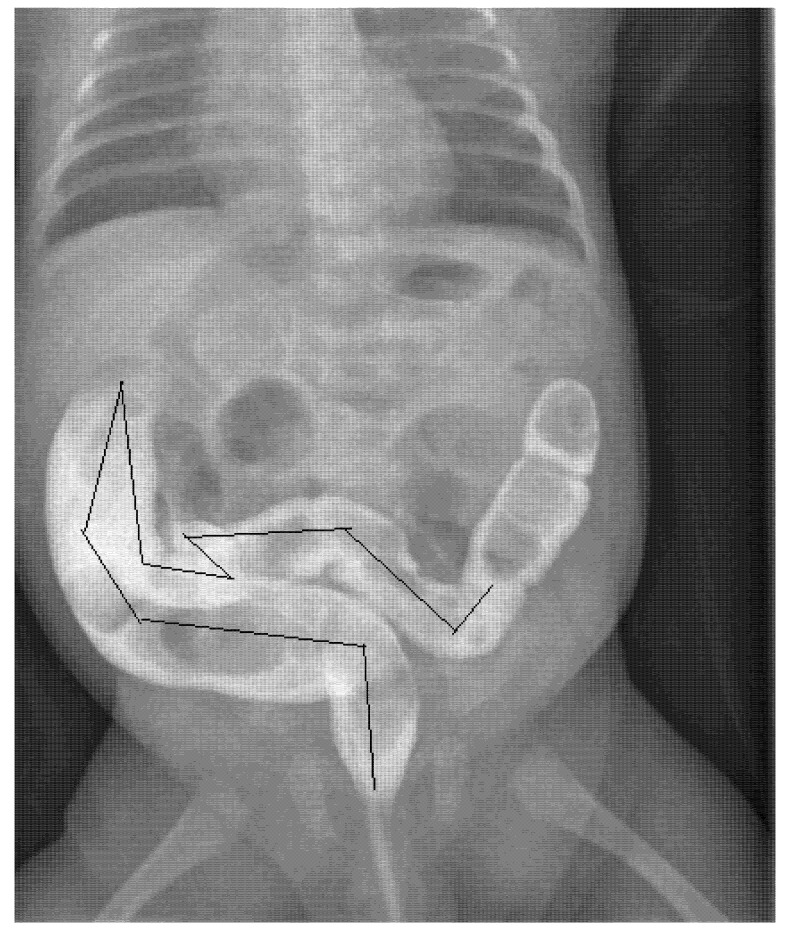
Image of CE of 2-months-old infant: the black line shows the rectosigmoid length.

**Table 1 children-09-01097-t001:** Comparison of calculated ratios with those reported by the literature as normal (age 1 year).

	Reference	Patients	*p*-Value
Cases	25	15	
Rectum diameter/L2	1.27	1.46	0.559
(IQ range)	(1.13–1.39)	(1.22–1.68)	
Sigmoid diameter/L2	1.05	1.20	0.047
(IQ range)	(0.98–1.10)	(0.91–1.44)	
Descending colon diameter/L2	1.15	1.01	0.041
(IQ range)	(1.04–1.29)	(0.82–1.44)	
Transverse colon diameter/L2	1.33	1.28	0.933
(IQ range)	(1.18–1.51)	(0.81–1.63)	
Ascending colon diameter/L2	1.33	1.24	0.032
(IQ range)	(1.22–1.5)	(1.08–1.36)	
Rectosigmoid length/L2	9.81	18.37	<0.001
(IQ range)	(7.98–11.70)	(13.69–26.29)

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
