# Peer review of "Can Infant Dyschezia Be a Suspect of Rectosigmoid Redundancy?"

_children, 2022, doi:10.3390/children9071097_

Round 1
Reviewer 1 Report
This is a study of dyschezia in infants associated with sigmoid elongatum. In this study, 24 infants with a suspected diagnosis of dyschezia are evaluated. 9 of these children showed aganglionosis (n=6) or constipation (n=3) after detailed diagnosis. The 15 remaining children were diagnosed with dyschezia, which is characterised by an agonising bowel movement. It is indeed a known phenomenon that parents present their children because, according to the parents, they have to squeeze hard and, in their opinion, torture themselves during defecation. This is also emphasised by the authors. In the first year of life, the frequency of defecation varies greatly, especially if the children are still breastfed. According to this study, 66% of the children were still fully (25%) or partially (42%) breastfed. In addition, the authors describe that the stool was soft or even liquid in most cases. This does not fit the diagnosis of "dyschezia". Further clinical specifications are missing. The details of possible complications, such as colonic volvulus, are given in the literature as casuistics, so that this risk must be considered very low. In addition, colonic volvulus occurs preferentially in the African population. There is no information or comment on this. The conclusion that a sigma elongatum, if this diagnosis actually exists, possibly plays a role, but should not be approached surgically, should be emphasised even more clearly. The authors do not comment on how far a bias could exist. Overall, however, the topic seems to me to be of interest to paediatricians and paediatric surgeons. Therefore, I recommend the revision of the manuscript. I also strongly recommend a revision by a native speaker.
Author Response
Dear Reviewer,
Thank you for your valuable suggestion.
We really appreciate your comment and we will try to improve my paper.
We have made the modification (red in the text) according to your suggestions
This is a study of dyschezia in infants associated with sigmoid elongatum. In this study, 24 infants with a suspected diagnosis of dyschezia are evaluated. 9 of these children showed aganglionosis (n=6) or constipation (n=3) after detailed diagnosis. The 15 remaining children were diagnosed with dyschezia, which is characterised by an agonising bowel movement. It is indeed a known phenomenon that parents present their children because, according to the parents, they have to squeeze hard and, in their opinion, torture themselves during defecation. This is also emphasised by the authors. In the first year of life, the frequency of defecation varies greatly, especially if the children are still breastfed. According to this study, 66% of the children were still fully (25%) or partially (42%) breastfed. In addition, the authors describe that the stool was soft or even liquid in most cases. This does not fit the diagnosis of "dyschezia". Further clinical specifications are missing.
- We are sorry, probably the exposure was not clear, the evaluation of the stool characteristics was made the day after admission, some children got intestinal softeners as described, but the stool characteristic in patients with infant dyschezia was 14 soft stools and one pasty stool, now we have specified it in the text.
The details of possible complications, such as colonic volvulus, are given in the literature as casuistic, so that this risk must be considered very low. In addition, colonic volvulus occurs preferentially in the African population. There is no information or comment on this. The conclusion that a sigma elongatum, if this diagnosis actually exists, possibly plays a role, but should not be approached surgically, should be emphasised even more clearly.
- You are right. Obviously the redundancy is not treated to prevent volvulus, but it creates a chronic and intractable constipation. We don't know how to treat it, more studies are needed. We have modified the discussion in this way: “We believe it is important to recognize the presence of a redundancy of the rectosig-moid colon, because this anomaly can rarely and especially in some ethnic groups pre-dispose to acute episodes of volvulus (12-17). Further studies are needed to understand the treatment of this abnormality. Having a longer part of the colon leads to constipation over the years, probably because the redundant pathway corresponds to a slow passage of stool (18 - 19); in our opinion only adequate medical therapy, associated with a correct in-take of fluids, can avoid severe forms of constipation in adults that some authors believe to be treated surgically (20) up to colectomy (21 - 22).”
The authors do not comment on how far a bias could exist.
- We added in the text: “In our case series, we did not assess all infant dyschezia present during the study period, but those that were referred to our center for suspected intestinal malformation, leading to select a group of patients with special features; in fact after our analysis the 15 infants had a sure diagnosis, as we ruled out other pathologies.”
Overall, however, the topic seems to me to be of interest to paediatricians and paediatric surgeons. Therefore, I recommend the revision of the manuscript.
I also strongly recommend a revision by a native speaker.
- We did a new English review of the paper and we showed the corrections in red.
Reviewer 2 Report
This manuscript describes a small cohort of infants presenting with stooling difficulties. Approximately one third were shown to have an organic condition, with 15 infants diagnosed with dyschaezia. The length of the rectosigmoid was assessed, with the findings suggesting longer length in the infants with dyschaezia
SPECIFIC COMMENTS
1. the work involved a very small group of infants, limiting the value arising
2. The assessment of the infants without an organic condition was cross-sectional: it would've been helpful to include a longitudinal assessment of outcome (not repeated imaging, just clinical outcome
3. Did all the infants undergo enema? In an infant diagnosed with hypothyroidism, there would not seem to be any indication for an invasive imaging procedure with radiation exposure
4. Did the rectosigmoid length correlate with the age of the infants with dyschaezia? or any other clinical outcomes
5. The ABSTRACT includes a number of errors of grammar and language usage: these should be corrected
6. The INTRO is one single long paragraph. Please revise with reformatting of the INTRO to enhance flow and readability
7. The METHODS is one single long paragraph: again please reformat into appropriate sections (using subheadings) to enhance readability
8. The indication for referral (for suspicion of surgical problems) is very broad and vague. Was there further definition of the referral indication?
9. The RESULTS should also be subdivided into sections with appropriate subheadings
10. Tables 1 and 2 are not required: this information can be easier started in the text of the RESULTS
11. The RESULTS should describe the characteristics of the infants with FD: age, sex, length of symptoms, growth parameters etc
12. Did the findings provided in the infants with FD differ in age or other clinical parameters? could the radiological findings be a secondary phenomenon?
13. The DISCUSSION is a suitable length.
14. References to the current work should be in the past tense (e.g. the aim of the study mentioned early on should say was and not is)
15. The conclusions at the end of the DISCUSSION refer to general facts (the natural history of FD) - they should refer only to the data arising in the current work
16. There are a number of awkward sentences and grammatical errors that need correction
Author Response
Dear Editor and Reviewer,
Thank you for your valuable suggestion.
We really appreciate your comment and we will try to improve my paper.
We have made the modification (red in the text) according to your suggestions
- The work involved a very small group of infants, limiting the value arising.
Dear Review, you are right, obviously the case series is not wide, but we have tried to compare it with the data of the literature because the data considered normal had a similar number of patients (some age groups 36 cases, other 4 cases).
Koppen IJ, Yacob D, Di Lorenzo C, et al. Assessing colonic anatomy normal values based on air contrast enemas in children younger than 6 years. Pediatr Radiol. 2017; 47: 306 – 1
- The assessment of the infants without an organic condition was cross-sectional: it would've been helpful to include a longitudinal assessment of outcome (not repeated imaging, just clinical outcome
We regret not having reported this data, but all the patients diagnosed with infant dyschezia have been referred to the pediatricians we have been in contact with for pediatric follow-up (we are a Pediatric Surgery Center).
We have added in materials and methods: “In these cases, the CE images were studied, then the patients followed a pediatric follow-up until one year of age.”
We added in results: “Patients diagnosed with infant dyschezia had in every case the follow-up until one year of age: all showed resolution of the problem and normal evacuation for age.”
Did all the infants undergo enema? In an infant diagnosed with hypothyroidism, there would not seem to be any indication for an invasive imaging procedure with radiation exposure
Of course, you are right, in the text it was written like this: “CE was performed in 23 cases (excluding the patient diagnosed with hypothyroidism on blood tests)”
- Did the rectosigmoid length correlate with the age of the infants with dyschezia? or any other clinical outcomes
We didn't find correlation with the age of infants with dyschezia
- The ABSTRACT includes a number of errors of grammar and language usage: these should be corrected
We did a new English review of the paper and we showed the corrections in red.
- The INTRO is one single long paragraph. Please revise with reformatting of the INTRO to enhance flow and readability
We have reformatted the INTRO respecting the indications of the journal.
- The METHODS is one single long paragraph: again please reformat into appropriate sections (using subheadings) to enhance readability
We have reformatted the INTRO respecting the indications of the journal.
- The indication for referral (for suspicion of surgical problems) is very broad and vague. Was there further definition of the referral indication?
We are a pediatric surgery unit of reference for gastroenterologists and pediatricians who suspect intestinal malformations. We have no further definitions, these specialists refer patients when they suspect an organic problem.
- The RESULTS should also be subdivided into sections with appropriate subheadings
We have reformatted the INTRO respecting the indications of the journal.
- Tables 1 and 2 are not required: this information can be easier started in the text of the RESULTS
You are right, we removed table 1 and 2
- The RESULTS should describe the characteristics of the infants with FD: age, sex, length of symptoms, growth parameters etc
We calculated these characteristics and added in the text: “At the end of the examinations the diagnosis of infant dyschezia was confirmed in 15 cases (9 male) who were recruited for the study. The mean age at presentation was 6 months (range: 2 - 9 months); in all patients the growth was regular for age, while the length of symptoms was different among patients, on average 5 months (range 1 - 7 months).”
- Did the findings provided in the infants with FD differ in age or other clinical parameters? could the radiological findings be a secondary phenomenon?
The only difference was the length of the symptoms: some were referred shortly after the onset of the disturbances, others after months, this is probably also linked to parental compliance. We added these sentence in the text
- The DISCUSSION is a suitable length.
- References to the current work should be in the past tense (e.g. the aim of the study mentioned early on should say was and not is)
We corrected it: “the authors noted that infant dyschezia was present even after 6 months of life and there-fore age of inclusion was recently been extended from 6 months to 9 months (2, 10).”
- The conclusions at the end of the DISCUSSION refer to general facts (the natural history of FD) - they should refer only to the data arising in the current work
We removed the general part.
- There are a number of awkward sentences and grammatical errors that need correction
We did a new English review of the paper and we showed the corrections in red.
Round 2
Reviewer 1 Report
After revision the manuscript has gained a lot and I think now it is of interest for readers of "Children". Nevertheless, some points remain to be addressed. Of the 15 patients with rectosigmoid redundancy length and width of the colonic parts were measured by 3 independent physicians. But it remains unclear whether intra- and/or interobserver differences were measured. The authors should comment on this, because it could rule out bias to some extent. But overall, I feel that after minot changes publication should be possible.
I recommend to check the English language by a native-speaker. I have the impression that the language and style are not quite right.
Reviewer 2 Report
Thank you for your revisions